# Do local and landscape context affect the attractiveness of flower gardens to bees?

**Devon S. Eldridge[1], Amani Khalil[1], John K. Moulton[2], Laura Russo[1]***

**1** Department of Ecology & Evolutionary Biology, University of Tennessee, Knoxville, TN, United States of America, **2** Department of Entomology & Plant Pathology, University of Tennessee, Knoxville, TN, United States of America

\* lrusso@utk.edu

## Abstract

Planting floral resources is a common strategy for increasing the abundance and diversity of beneficial flower-visiting insects in human-modified systems. However, the context of the local area and surrounding landscape may affect the attractiveness of these floral resource provisioning plots. We compared the relative effects of local floral resources and surrounding urban land-use on the abundance of bees on flowering plants in common gardens in eastern Tennessee, USA. We planted four types of common garden plots at each of five different landscapes representing a variety of surrounding land use: 1) Urban Garden, 2) Forage Grassland, 3) Mixed Agriculture, 4) Forest, and 5) Organic Farm. Each common garden plot type had a fixed plant community representing one of three plant families (Asteraceae, Fabaceae, Lamiaceae) or a mix of all three, and all four common gardens were replicated at all the sites. We concurrently sampled bees in the garden plots and in a 50 m radius (local area) around the garden plots. We found that the size of the floral display (i.e. the visual display size of flowers) and diversity of flowers in the local area did not affect bee abundance or species richness in the garden plots. Although there was a significant positive association between developed land use in a 2 km radius and bee abundance in the gardens, the effect was small, and there was no relationship between land use and bee abundance or species richness in the local area. There were significant differences in the composition of the bee community between the local area and garden plots, but the largest determinants of bee community composition and species richness in the gardens were floral display size and variation in the garden plant species in bloom. This finding is promising for anyone wishing to promote pollinator populations by providing more floral resources.

## 1. Introduction

The conservation of beneficial biodiversity, such as pollinating insects, is a topic of significant concern, particularly because these organisms contribute to human health and well-being. For example, as the human population grows, so does the global production of food, both in terms of the extent and intensity of agricultural production [1]. Many crop species rely on insect pollination, and land-use intensification has been shown to reduce both pollinating insect species

**Data Availability Statement:** All relevant data are within the manuscript and its Supporting Information files.

**Funding:** Funding was provided by Bayer's Feed-A-Bee grant to LR in 2019. The funders had no role in

study design, data collection and analysis, decision to publish, or preparation of the manuscript.

**Competing interests:** The authors have declared that no competing interests exist.

richness and abundance [2,3]. At the same time, insect biodiversity is key to pollination services [4]; for example, while native bees can provide sufficient pollination services on watermelon farms, continued agricultural intensification drastically decreases unmanaged pollination services [5–7]. In fact, land use changes have been implicated as a primary driver of insect declines worldwide [8]. Thus, better understanding the role of land-use change and how to mitigate its negative impacts on beneficial biodiversity is an important topic of research.

Another area of increased land-use change lies in developing, or urbanizing, landscapes. The extent of developed landscapes is accelerating globally and although they are considered by some to be refuges for bees [9–11], urban areas can have negative effects on pollinating insects [12,13]. Urban land-use is unique in the way it alters landscapes; cities are considered hotspots of biological invasion [14], including for invasive bees [15,16], and significantly alter abiotic factors by increasing temperatures and the extent of impervious surfaces [17], which then reduce the soil available for ground-nesting pollinators and floral resources [18]. Both agricultural and urban land-use have repeatedly been shown to affect pollinating insects [19,20]. For example, agricultural land use was shown to decrease phylodiversity of bee communities [3], and urban land use was shown to have a negative effect on gene flow in *Bombus vosnesenskii* [21] and to change the microbiome in *Ceratina calcarata* [22]. While the negative impacts of agricultural land use seem to depend on the intensity and extent of the management [23], urban land use does not always have clear negative effects on pollinating insects, and the impact can depend on taxa and traits [24,25]. It is possible that local floral resources, such as urban gardens, may be mediating the impact of urban land use [26]. Moreover, bees are central place foragers, meaning that they forage within a given area and return to provision their nest, making them more vulnerable to certain spatial stressors [27]. This may mean that the structure of habitat patches within landscapes may also affect bee abundance and diversity. The foraging range of bees, and therefore the potential for the broader landscape to influence their foraging decisions, is also affected by their adult body size [28], though there is a significant difference between the potential and realized foraging distances of bees [29].

As a way to ameliorate the impact of such intensive land-use on beneficial arthropods, those interested in protecting pollinating insects recommend floral resource provisioning strips [9,30–32]. These strips of floral resources, essentially flower gardens, have been shown to increase pollinator abundance and diversity in intensively managed systems [30,33,34], but their attractiveness to pollinators also seems to vary with landscape context [35–37]. Attractiveness is a concept that dictates the foraging preferences of pollinating insects. It is usually defined in relative terms: a plant is attractive relative to other plants if it receives a greater abundance of insect visitors given the size of its floral display, or the visual component of bloom [38]. The attractiveness of flowering gardens to pollinators may be affected by local factors, such as floral resources in the immediate area around the garden [35]. For example, at the local scale, the abundance of visitors to any given plant species might be affected by neighboring flowering plants [39]. There is evidence for both a magnet effect, where having an attractive neighbor increases visitation to a plant [40], and a competitive effect, where having an attractive neighbor decreases visitation to a plant [41]. These two effects may in part be moderated by the availability of resources or environmental filtering at a broader scale [37] and a recent review demonstrated prevalent scale-dependence in pollinator-mediated facilitation between plant species [42].

Environmental filtering is the concept that the species found in any given habitat are a subset of the species in the surrounding community [43]. Thus, the surrounding landscape may determine to some extent the pollinator species available to visit a given plant, while the local area may determine the relative attractiveness, or the relative abundance of insect visitors to

the flowers [38]. This concept has been applied to other urban green space communities; for example, the composition of carabid communities in urban green spaces was found to be strongly structured by environmental filtering [43].

Our goal was to determine whether the attractiveness of flowering garden plots was affected by local or landscape factors, or an interaction between them. We established four different types of common garden plots at each of five different sites. Three of the common garden plots established at each site included six species each of the plant families Lamiaceae, Fabaceae, and Asteraceae, while the fourth included two species from each of those three plant families. The study sites varied in both the availability of local flowering resources and the surrounding land-use. Our hypotheses were that: 1) the abundance and diversity of bees visiting flowers in the garden plots would be mostly driven by the abundance and diversity of local flowering resources, 2) land-use at a broader scale would interact with local floral availability to affect bee abundance and richness in the gardens, and 3) the bee community in the gardens would be a subset of the surrounding bee community.

## 2. Materials and methods

### Experimental design

We established garden plots with a fixed community of native perennial plants in eastern Tennessee, USA. Each of the gardens contained four individuals of each of six perennial wildflower species native to Tennessee. In each of five sites, we planted four 3m x 2m garden plots: 1) six species of the plant family Asteraceae (A), 2) six species of Fabaceae (F), 3) six species of Lamiaceae (L), and 4) a mixed garden plot two species of each of the aforementioned families (mixed plot, M) (Fig 1). These three plant families were selected due to the range of floral resources they provide, from primarily nectar-based resources in the Lamiaceae, to protein-rich pollen in the Fabaceae [44]. The plants also differed in their floral morphology and these traits did interact to affect pollinator visitation [45]. At each site, the garden plots were separated between 15m – 50m. Sites were all property owned by the University of Tennessee (Research and Education Centers, or RECs). The distance between sites ranged from 1 km to 106 km. We chose the plant composition (a total of 18 species) based on commercially available, insect-pollinated plant species native to the region and sourced them from a local native plant nursery (Overhill Gardens, Vonore, TN, S1 Table in S2 File).

All of the gardens were established following the same methodology at five separate sites: 1) Urban Garden, 2) Forage Grassland, 3) Mixed Agriculture, 4) Forest, and 5) Organic Farm. These sites were selected because they represented a variety of local and landscape types, with land-use at a 2 km radius varying from primarily urban (Urban Garden) to primarily agricultural (Forage Grassland). The community of flowering plants immediately surrounding the gardens varied from frequently mown (Forage Grassland), to a high diversity of native (Forest) and ornamental (Urban Garden) flowering beds. Both the Mixed Agriculture and Organic Farm sites comprised a mixed community of taller grasses and some common non-native flowering weeds, that were mown twice a summer, alongside a mixture of research crops including blueberries, apples, and switchgrass (Mixed Agriculture) and lettuce, squash, and mint (Organic Farm). The fixed community of plants within the gardens allowed us to isolate the effects of local and landscape factors on changes in the flower-visiting insect communities.

### Garden plot surveys

Each garden plot was visited weekly from July 13th–August 17th, 2020, or five complete rounds of sampling each plot in the study. This represented the peak bloom period of the research plants in our study plots. While some of the plant species in the plots bloomed outside

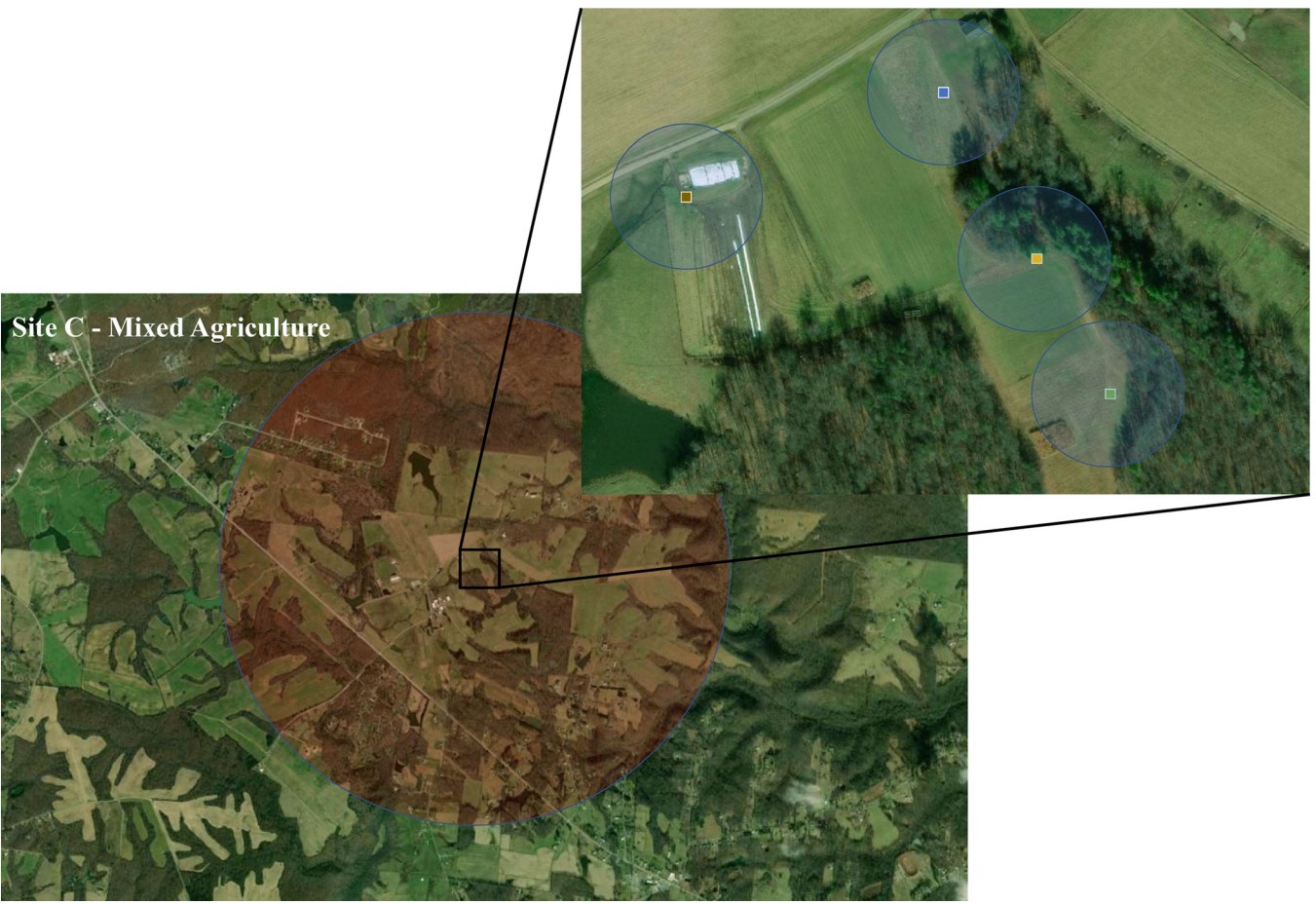

**Fig 1. A map of one of the study sites illustrating the sampling design.** The larger map illustrates the 2 km radius, from which we assessed land use. The inset show the four research gardens represented by squares (Lamiaceae = blue, Fabaceae = yellow, Asteraceae = green, Mixed = brown), which are surrounded by the 50-meter local sampling area. Base map from USGS EROS Earth Explorer (public domain: https://earthexplorer.usgs.gov/).

this period, it represented the greatest proportion of co-flowering plants, and largest floral display of plants in the plots. The amount of time spent sampling in each garden varied depending on the number of plant species in flower (5 min per plant species in bloom, or up to 30 min per garden). For the gardens, floral diversity was represented by the number of plant species in bloom during the survey.

We collected all insects that contacted the reproductive parts of the inflorescences within the garden plots during standardized surveys. These weekly surveys involved sampling each plant species in bloom in each plot at a given site for five minutes, using a hand-held vacuum aspirator from BioQuip (Rancho Dominquez, CA). During each collection event, we also counted the number of inflorescences of each plant species and used the number of inflorescences per plot as a measure of overall floral display. There were three different plant families in the plots, with different floral morphologies. The Asteraceae all had composite inflorescences, or flowerheads, and each flowerhead was counted. For Fabaceae that presented large individual flowers (all but *Amorpha herbacea*), we counted these, and for *Amorpha herbacea*, we counted racemes of flowers as inflorescences. For the Lamiaceae that presented large individual flowers (all except *Blephilia subnuda*, *Lycopus virginicus*, and *Pycnanthemum muticum*), we counted these, while for *B. subnuda*, *L. virginicus*, and *P. muticum*, we counted the

composite-like flowerheads as inflorescences. We kept track of this to determine how the background floral display within the plot affected its attractiveness to pollinating insects [46–48].

## Local survey

We paired the garden plot surveys with surveys in the local area immediately around each plot, because we were interested in whether the local flowering plants affected visitation to the research plots. These local surveys were paired in time (July 13th–August 17th 2020) and space with the plots. An insect net was used to capture insects that contacted the reproductive parts of flowers within a 50m radius of each research plot once a week, for a total of five complete surveys (Fig 1). These local surveys were 10 minutes each, allowing the collector to walk the entirety of the area within the 50m radius, and the timer was paused after each insect was collected, while it was being transferred to a collection vial. Honeybees (*Apis mellifera* L) and carpenter bees (*Xylocopa virginica* L) were counted and recorded but not captured, as they could be identified on sight.

Flowering plant diversity (floral diversity) and floral display were surveyed before each pollinator survey was performed. In these local surveys, we did not identify the flowering plant species, but rather were interested in rapidly assessing how many different flower types were available (diversity) and how abundant they were (floral display), because our main goal was determining whether the abundance and diversity of floral resources around the gardens had any effect on bee visitation to the fixed plant communities within the gardens. The same surveyor ranked diversity and floral display on all of the local surveys, to ensure that the rankings would be consistent. We used a visual ranking method within the 50m radius around each research plot in the following manner. First, we ranked the floral diversity on a scale from 1–10, where 1 was the least diverse survey area (i.e. a single plant type in bloom), and 10 was the most diverse survey area (a variety of colors and floral morphologies in bloom). We used a mowed lawn as an example of a 0 on the floral diversity scale, and a botanical garden in bloom as a 10 on the floral diversity scale. Next, we ranked floral display on a scale from 1–10, where 1 was the smallest, and 10 was the largest floral display for the survey areas. Similarly, a mown lawn with no flowers was a 0 on this scale, and an apple orchard in full bloom was our example of a 10 on this scale. Our local surveys were ranked on these two scales by the same surveyor before each pollinator collection event. Such ranking methods have been shown to be consistent with other vegetation measures and respond consistently to treatments [49,50]. We chose this method of assessment over the Braun-Blanquet method because we wanted a rapid visual assessment of the entire local area around the gardens, rather than quadrat subsamples [51]. These measures allowed us to qualitatively compare the survey areas to one another.

## Insect processing

Insects collected from both garden and local surveys were frozen, then pinned, labeled, identified, and databased. We focused our analysis on bee specimens (members of the superfamily Apoidea). Bees were identified to genus and species where possible, using the Discover Life interactive key [52]. Bee identifications were then verified by Sam Droege (USGS). Specimens are vouchered at the University of Tennessee.

## Landscape analysis

To classify the landscape around our research sites, we used ArcGIS Pro 2.6 and the US National Land Cover Database (NLCD) (https://www.mrlc.gov/), with a 30 m resolution [53]. We classified land cover at 2000 m around the center of each site (Fig 1). Our goal in using this buffer radius was to determine how the broader landscape affected bee diversity and abundance in the gardens [54,55]. In our analysis, we detected 14 land cover classifications that we

aggregated into four general land-use types: water (Open Water), developed (Developed High Intensity, Developed Low Intensity, Developed Medium Intensity, Developed Open Space), agriculture (Cultivated Crops, Grassland/Herbaceous, Pasture/Hay), and semi-natural (Deciduous Forest, Emergent Herbaceous Wetlands, Evergreen Forest, Mixed Forest, Shrub/Scrub, Woody Wetlands) (S2 Table in S2 File). The NLCD defined developed land as constructed materials and impervious surfaces such as commercial and residential housing, roadways, and lawn grasses. Areas classified as water included open water and areas with minimal soil and vegetation. Semi-natural land use included different forest types, wetlands, shrubland, and non-grassland herbaceous land cover. Agricultural land use comprised of pastureland, cultivated crops, or grasslands. We hypothesized that landscape context would interact with local floral resources to have an effect on bee diversity or abundance; specifically, that landscapes that increased bee diversity and abundance in the local surveys would also increase bee diversity and abundance in the garden plots. We selected 2000 m to provide a strong contrast with the local surveys; however, for completeness, we reran these analyses with land-use buffers calculated at 500 m and 1000 m.

## Data analysis

All data analyses were conducted in R version 4.3.2 [56,57]. First, we compared the bee collections among the sites and surveys (garden or local). We used feature scaling to standardize the measures of floral display and diversity in the gardens and local area on a scale of 0–1. We used rescaling (aka min-max normalization) here, following the equation: $x' = \frac{x - \min(x)}{\max(x) - \min(x)}$. We compared the average number of bees collected per minute sampling, and the relationship between bee abundance and the scaled floral display for both survey types. We used a rarefaction analysis (function *iNEXT* [58]) to test for differences in bee diversity among the sites and surveys. This function uses sample-size based integrations to calculate Hill numbers, or the effective number of species, to quantify the species diversity of an assemblage (58). We report the results for Hill numbers $q = 0$ (species richness) and $q = 1$ (Shannon diversity). Shannon diversity incorporates information on the relative abundances of different species, or species evenness, as a way to interpret the effective number of species in a community. Shannon diversity is often included as a way of measuring diversity beyond just the count of species (i.e. species richness). *iNEXT* has the added benefit of accounting for sample size, which is helpful in comparing communities of different sizes or where sampling effort is uneven. Where honeybees were especially prevalent, we ran the rarefaction analysis with and without honeybees. This species was of special interest because it is a non-native species that can potentially compete with wild bees for floral resources [16,59,60], but it is unlikely to be affected by the same environmental factors because honeybees are often kept in managed hives. For example, honeybee abundance may be positively associated with urban or agricultural beekeeping [53]. We also tested for sampling completeness of both survey types using *iNEXT*.

We built four separate models for bee abundance and species richness in the garden and local surveys. To test our first hypothesis, we then used generalized linear mixed effects models (GLMMs) using the function *glmmTMB* in the package "glmmTMB" [61] to test the effect of the scaled floral display and floral diversity in the garden and local area, and developed land use at a 2 km radius on bee abundance and species richness in the gardens, with sampling round nested in plot as the random effect. Because the floral display and floral diversity were correlated, we ran two sets of models for each response, one with display and one with diversity. We then used the function *anova* to compare the two models, and selected the model with the lower AIC value to include. We also used GLMMs to test whether the abundance and species richness of bees in the area around the gardens was affected by land use or local floral display

and floral diversity. To test our second hypothesis about whether land use interacted with local floral display and floral diversity, we also tested for interactions between the land use effect and the other fixed effects. We removed non-significant interaction terms from the final models. For all count-based models, we used a negative binomial distribution due to overdispersion.

We used non-metric multidimensional scaling (NMDS) to compare the bee community composition among the sites and surveys [62]. NMDS is an ordination technique that allows for visualization of multivariate responses to treatments. In our case, we were looking for overlap, or non-overlap in the community structure of the bees. We used the function *anosim* in the package "vegan" [62] to test for significant differences in the bee community composition. We used 99 permutations and Bray-Curtis dissimilarity in this test. To determine which species were driving differences, we used the *multipatt* function in the package "indicspecies" [63] when the groups differed significantly.

## 3. Results

### Garden plot surveys

We collected 1,470 specimens during 20.83 hours of sampling in the garden plots [53]. Most of the specimens (1,186 or 81%) were bees (Halictidae, Megachilidae, and Apidae) (S1A Fig in S2 File), of which 137 (9%) were honeybees (*Apis mellifera*). We were able to identify all specimens to species except for 157 *Lasioglossum* males and five *Ceratina* females, which were left at the genus level (11% of all specimens collected in the gardens). A total of 44 bee species were identified from the garden collections. Halictidae (791 specimens) was the most abundant bee family collected within the gardens during the sampling period. Apidae followed as the second most abundant bee family with 356 specimens. The largest number of specimens were collected from plots at the Forage Grassland (379) and the Organic Farm (318).

On average, 2.4 ± 0.13 (standard error) plant species were in bloom per plot during the surveys. We observed an average of 369.21 ± 39.8 inflorescences per plot per survey. We used feature scaling to standardize these measures to a 0–1 scale to compare to the floral diversity and floral display of the local surveys.

### Local surveys

We collected or observed 3,324 flower-visiting insects during 16.67 hours of sampling the local areas surrounding the plots. Most of the specimens (2,917 or 88%) were bees (Halictidae, Megachilidae, Colletidae, Andrenidae, and Apidae) (S1B Fig in S2 File); 1,380 (42%) were *A. mellifera* (S1 Fig in S2 File). We were able to identify all specimens to the species level except for 92 *Lasioglossum* males and four *Ceratina* females (3% of the specimens collected in local surveys). We identified 52 species of bees (S3 Table in S2 File). We collected six species of bees not found in the plot surveys: *Melissodes communis* Cresson, *Xenoglossa* (*Peponapis*) *pruinosa* (Say), *Hylaeus leptocephalus* (Morawitz), *Lasioglossum fattigi* (Mitchell), *Lasioglossum simplex* (Robertson), and *Hoplitis producta* Cresson. *Lasioglossum fattigi* and *L. simplex* were both new occurrence records for the state of Tennessee (*pers. comm.*, S. Droege, J. Ascher).

The average ranked floral display in the local survey was 4.25 ± 0.27 (standard error) and the average ranked floral diversity was 4.17 ± 0.29. We used feature scaling to standardize these ranked floral display and floral diversity measures from 0–1.

### Comparing garden and local surveys

The Mixed Agriculture site had the highest bee abundance among the garden surveys, while the Urban Garden site had the highest abundance among the local surveys (Fig 2A). The local

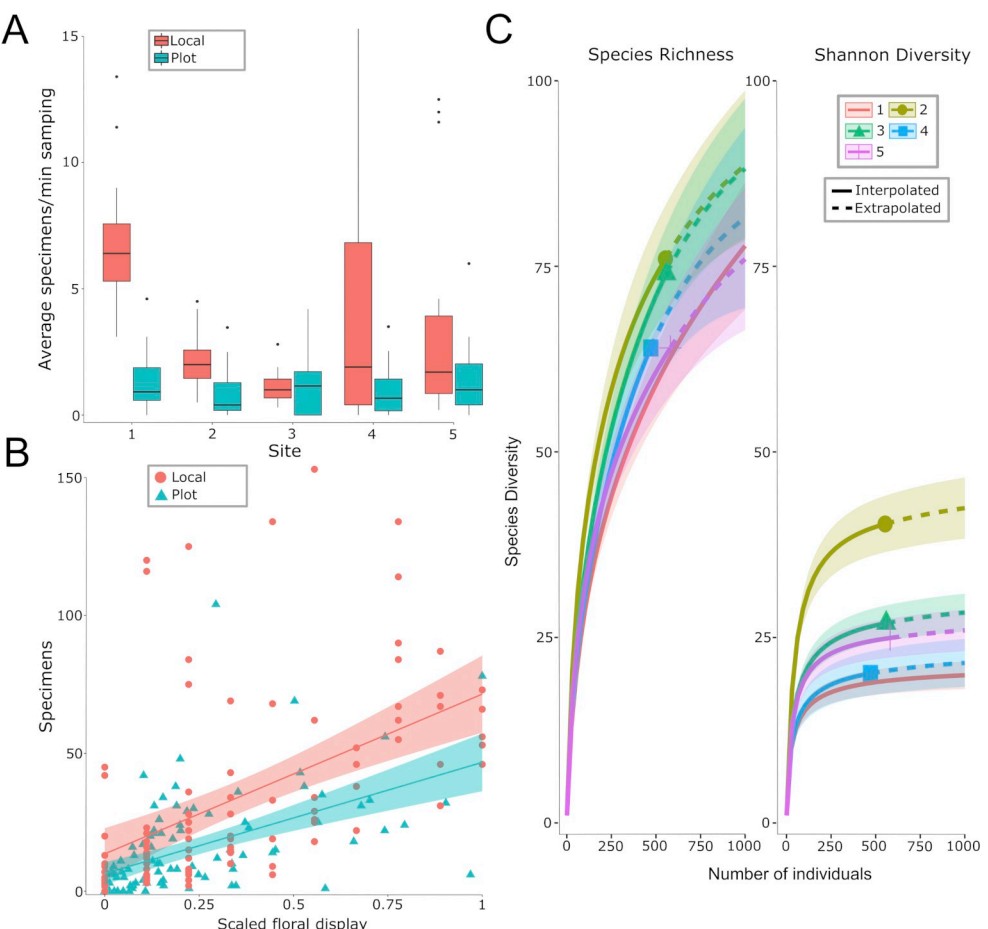

**Fig 2.** A) Box and whisker plots of the average number of bees collected per minute sampled at all five sites (1: Urban Garden, 2: Forage Grassland, 3: Mixed Agriculture, 4: Forest, and 5: Organic Farm) for the landscape (red) and plot (blue) surveys. B) Number of bees collected per sample compared to the scaled floral display of the local (red) and garden (blue) surveys. The lines are drawn with the function "geom_smooth(method = lm)" in the package *ggplot2* and the shaded area around each line indicates the 95% confidence interval. C) Rarefaction analysis (excluding honeybees) showing bee species richness and Shannon diversity at the five sites (1 = red, 2 = yellow, 3 = green, 4 = blue, 5 = purple). Interpolated data are represented with a solid line and extrapolated data are represented with a dashed line. The shaded area around each line is the 95% confidence interval.

surveys found a higher bee abundance per minute spent sampling than the garden plots at four of the five sites (Fig 2A). The scaled floral display had a significant positive effect on bee abundance in both local and garden surveys (Fig 2B). With honeybees included in the rarefaction analysis, the Forage Grassland and Mixed Agriculture sites both had significantly higher bee Shannon diversity than the other sites, but all the sites had overlapping confidence intervals for bee species richness (S2A Fig in S2 File, S4 Table in S2 File) and when we excluded honeybees, only the Forage Grassland had higher Shannon diversity than the other sites. We also compared the local and garden surveys (S2B and S2C Fig in S2 File) and found that they did not differ in bee species richness, but that the gardens had a higher Shannon diversity than the local surveys due to a greater evenness. This difference was probably driven by the dominance of honeybees in the local surveys because, after removing honeybees from the rarefaction analysis, we found that the local surveys had a higher Shannon diversity than the garden plots (S2C Fig in S2 File). Our sample coverage was above 95% for all the sites we sampled, and above 98% for both survey types (S4 Table in S2 File, S2D and S2E Fig in S2 File).

**Table 1. Results from generalized linear mixed effects models (GLMMs) for bee abundance and species richness in the gardens and local area.** Significant effects are bolded. Due to overdispersion, we used negative binomial models for the count-based responses.

| Response | Fixed effects | Contrasts | Family | Random effects | Observations | Estimate | z value | P value |
|---|---|---|---|---|---|---|---|---|
| Garden Bee Abundance | **Garden Floral Species Richness** | continuous | Negative Binomial | Round\|Plot | 100 obs, 4 plot types, 5 rounds | **3.84** | 6.03 | **<0.001** |
| | Local Floral Diversity | | | | | -1.12 | -1.71 | 0.09 |
| | **Development (2km)** | | | | | **0.02** | 2.07 | **0.04** |
| Local Bee Abundance | **Local Floral Display** | continuous | Negative Binomial | Round\|Plot | 100 obs, 4 plot types, 5 rounds | 1.29 | 2.09 | **0.04** |
| | Development (2km) | | | | | 0.01 | 1.66 | 0.1 |
| Garden Bee Species Richness | **Garden Floral Display** | continuous | Negative Binomial | Round\|Plot | 100 obs, 4 plot types, 5 rounds | **1.59** | 4.86 | **<0.001** |
| | Local Floral Display | | | | | -0.4 | -1.01 | 0.31 |
| | Development (2km) | | | | | 0.01 | 1.94 | 0.05 |
| Local Bee Species Richness | **Local Floral Display** | continuous | Negative Binomial | Round\|Plot | 100 obs, 4 plot types, 5 rounds | **0.89** | 3.36 | **<0.001** |
| | Development (2km) | | | | | 0.002 | 0.53 | 0.6 |

We did not find any significant interactions between developed land use at a 2 km radius and any of the other fixed effects and thus removed the interaction terms from the models. The best model for bee abundance in the gardens included the floral diversity in the local area and gardens, along with development. Bee abundance in the gardens was significantly associated with the flower species richness in the gardens (i.e. the number of plant species in bloom at the time of the sample), along with developed land use at a 2 km radius (Table 1, Fig 3A and 3B). However, the effect of development was much lower in magnitude than the effect of the floral richness in the gardens. Bee abundance in the gardens was not associated with local floral diversity. The best model for bee species richness in the gardens included the floral display in

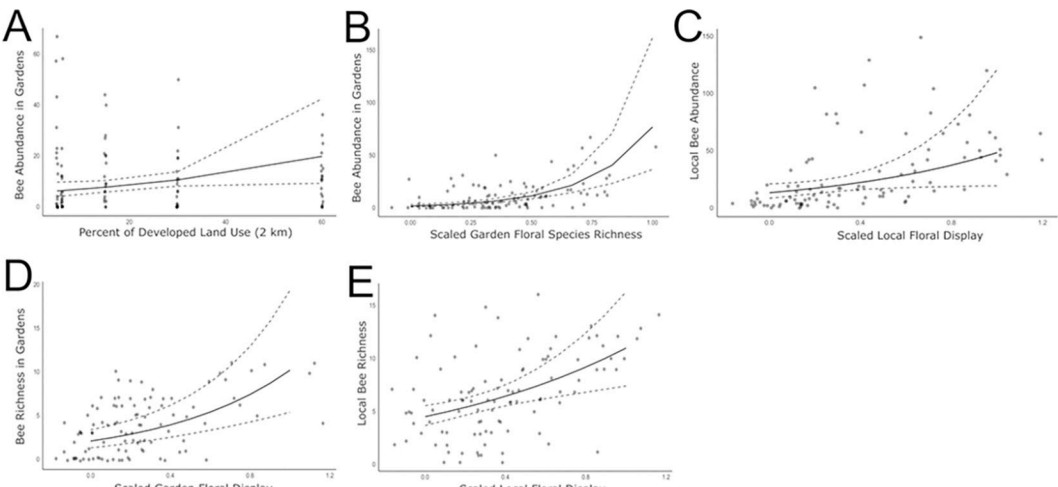

**Fig 3. Significant relationships between bee abundance and species richness in the gardens and local area and fixed effects in the models.** We found that bee abundance in the gardens was driven most strongly by the developed land use in a 2 km radius (A) and floral species richness in the gardens (B). Bee richness in the gardens was driven mostly by the floral display in the gardens (D). The local bee abundance (C) and local bee richness (E) were both driven by the local floral display. The points on the graph represent raw data and the solid lines are predicted relationships from the models. The dashed lines around the solid lines represent the 95% confidence intervals.

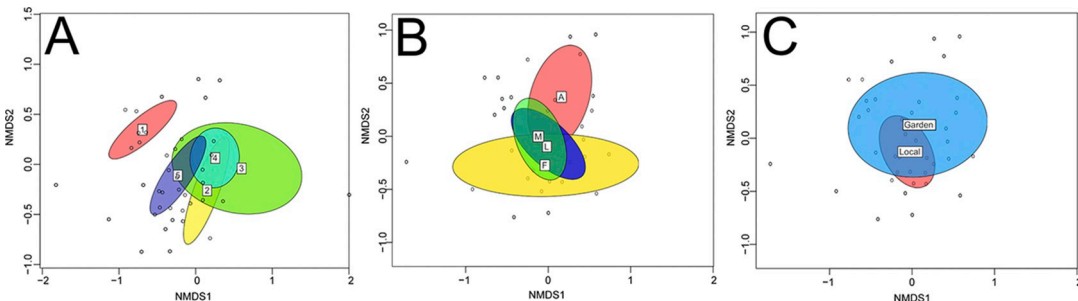

**Fig 4.** Non-metric multidimensional scaling (NMDS) ordination plots of the different sites (A), plot types (B), and surveys (C). Overlap in the shapes indicates overlap in the communities in the sites or surveys. Color indicates landscape (A): Urban Garden (red), Forage Grassland (yellow), Mixed Agriculture (green), Forest (light blue), and Organic Farm (dark blue); plot type: Lamiaceae (dark blue), Fabaceae (yellow), Asteraceae (red), and Mixed (green); and survey type: garden (blue) and local (red).

the gardens and local area, along with developed land use at a 2 km radius. Bee species richness in the gardens was significantly associated with floral richness in the gardens, but not local floral diversity or development (Table 1, Fig 3D).

The best model for bee abundance in the local surveys included the local floral display and developed land use at a 2 km radius. Bee abundance in the local surveys was significantly associated with local floral display, but not development (Table 1, Fig 3C). The best model for bee species richness in the local surveys included local floral display and developed land use at a 2 km radius. Bee species richness in the local surveys was only associated with local floral display (Table 1, Fig 3E). We reran these models with the land-use buffer calculated at 500 m and 1000 m, but the results did not differ qualitatively from the original analyses (S5 Table in S2 File).

The NMDS ordination plots (Fig 4, stress = 0.196) showed community overlap between the sites, plot types, and surveys. All three groups differed significantly from one another, but with low R values, suggesting some overlap in community composition (sites R = 0.19, p = 0.01; plot types R = 0.08, p = 0.04, survey types R = 0.19, p = 0.01). For this analysis, R values close to 1 would suggest complete dissimilarity in groups, while values close to zero would suggest community overlap. The indicator species analysis identified bee species that were found more often at a given site or plot type. This analysis suggested that the greatest variation in bee species composition was at the site level (S6 Table in S2 File). There were many bee species found significantly more often at the Urban Garden site; these significantly associated bee species were *Bombus impatiens*, *B. pensylvanicus*, *B. griseocollis*, *Megachile pusilla*, *M. rotundata*, *M. mendica*, *Xylocopa virginica*, *Lasioglossum apocyni*, *Anthidium manicatum*, and *Agapostemon virescens*. *Megachile rotundata* and *A. manicatum* are both non-native bee species often associated with urban habitat [14,64]. The bee species found to be associated with the Organic Farm site were *Lasioglossum zephyrum*, *L. admirandum*, *L. tegulare*. Only one bee species was found more often at the Forage Grassland site (*Ceratina dupla*), and the Forest site (*Lasioglossum lustrans*) (S6 Table in S2 File). For the plot types, the only significantly associated bee species was *Halictus ligatus/poeyi*, which was significantly associated with the Asteraceae plots (S6 Table in S2 File). Between the two survey methods, there were seven species significantly associated with the local surveys (*Apis mellifera*, *Lasioglossum hitchensi*, *L. callidum*, *L. imitatum*, *L. trigeminum*, *L. apocyni*, and *Calliopsis andreniformis*) and none associated with the gardens (S6 Table in S2 File).

## 4. Discussion

In our study of the relative effects of local and landscape characteristics on the attractiveness of flower gardens to visiting bees, we found strong positive relationships between floral display

and bee abundance and species richness, but also that the effect was mostly constrained to the sample area, whether it be the common garden itself or the local area. Similar relationships have been previously documented in agricultural ecosystems [65,66] and strong background effects of floral display on pollinator visitation are well-documented [48,67]. We observed the highest local floral diversity and floral display in the site managed as a public garden (Urban Garden site). This site was well manicured, with many flower beds for display. At the same time, this high floral display and diversity in the local area did not significantly increase the abundance or species richness of the bees in our garden plots. Our first hypothesis, that local floral display and diversity would be the primary driver of bee abundance and species richness in the garden plots, was therefore not supported by our results. This contrasts with other studies that have shown that the surrounding floral resources and landscape context can significantly affect bee abundance and diversity in wildflower gardens [35,68].

Our second hypothesis was that the surrounding landscape would interact with local floral resources to drive bee abundance and species richness in the gardens. However, we found no significant interactions between our land use variable and the other factors in our analysis. While developed land use at a 2 km radius did have a significant positive effect on bee abundance in the gardens, the effect size was orders of magnitude smaller than the effect of the floral display in the gardens themselves. Moreover, developed land use had no effect on the bee abundance or species richness in the local surveys. Thus, our second hypothesis was also not supported. Overall, local factors seemed to be the major drivers of the bee community, at least in this study. It is possible that other land-uses, such as agricultural land or natural areas, may have had an undetected influence on the pollinator communities in these systems. Further, our sites were in patchy landscapes and this patchy, heterogenous habitat may influence bee species composition and abundance [69,70]. Measuring landscape heterogeneity and patch structure in the landscape might show different patterns.

It is also possible that more extreme landscapes would have shown stronger effects. Our sites varied between roughly 5% and 60% developed land-use at a 2 km radius, whereas one previous study found non-linear effects on homogenization in Lepidoptera above 60% agricultural land-use [71]. Our findings might therefore not be representative of more homogenous landscapes. However, even if this were true, we would at least expect to see some difference between the two extreme ends of our gradient. Instead, we saw a mild positive effect of developed land use on overall bee abundance and species richness in the gardens. Previous studies similarly found multiple non-linearities and significant variation in patterns of community similarity [72].

Our third hypothesis was that the bee community in the research gardens would be a subset of the local bee community. Overall, the sites and surveys had relatively similar total bee species richness, but the evenness of the species varied, leading to a greater difference in the Shannon diversity. For example, although there were not significant differences in the total bee species richness of the two surveys, the garden survey had a higher Shannon diversity until we removed honeybees from the analysis. Proportionally, far fewer honeybees were collected in the gardens (~9%) than in the surrounding local surveys (~53%), which may have been due to the plant species composition of the garden plots relative to the local area. Eusocial bee species like the honeybee may be capable of competitively excluding solitary bee species [59,60,73], but may also exhibit distinct foraging preferences [74] and may not be affected by the same land-use factors [16,53]. Although on a per minute sampling basis, the local surveys collected a higher abundance of bees, from a bee community composition perspective, the differences among sites, plot types, and survey types were relatively minor. Similarly, though these groups differed significantly, there was substantial overlap in the community composition. We were able to identify significant indicator species among the different groups, and found that the

Urban Garden and the local surveys had the largest number of significantly associated species, while there were no bee species significantly associated with the research gardens. From these results, we find mixed support for the hypothesis that the bees in the garden plots were a subset of the surrounding bee community.

## 5. Conclusions

Our study provides promising findings for anyone that wishes to promote diverse pollinator communities. Along with other research, this study supports strategies for promoting pollinators by increasing floral diversity and floral display [75], especially when these plants provide a diverse range of floral resources [45]. Our findings did not show support for our hypotheses that bee abundance and species richness in experimental gardens would be driven by a mixture of local and landscape effects. Instead, we saw that these gardens attracted a diverse bee community largely through the size of their floral display and the identity of the blooming plants within them. We found minor effects of developed land use but no large effects across a range of land use. We show that local floral resources are stronger predictors of bee abundance than surrounding land-use and that managed landscapes can maintain high abundances of bees. Whether undisturbed or managed, landowners have the power to promote their local pollinator communities by conserving existing natural patches or increasing the availability of floral resources.

## Supporting information

**S1 File. Supporting information appendix: Data collected for the analyses in this study.** (XLSX)

**S2 File. Supporting information appendix 2: Supplemental tables and figures, including six additional tables and two additional figures.** (DOCX)

## Acknowledgments

We would like to thank the University of Tennessee (UT), the UT Institute of Agriculture, the UT Gardens, and the UT Ag Research and Education Centers, including the Plateau AgResearch Center, the Organic Crops Unit, and the Forest Research and Education Center, for allowing us to do research on their lands, especially K. Hoyt, B. Simpson, W. Hitch, H. Jones, J. Newburn, and W. Lively. We would also like to thank D. Matheson, S. Collins, and A. Murray for field assistance and N. Oldham for assisting with hoverfly identification. Thanks also go to S. Droege for lending his expertise in bee identification. Thank you to the Department of Ecology & Evolutionary Biology at the University of Tennessee for the support of this project.

## Author Contributions

**Conceptualization:** Devon S. Eldridge, Amani Khalil, Laura Russo.

**Data curation:** Devon S. Eldridge, Amani Khalil, John K. Moulton, Laura Russo.

**Formal analysis:** Devon S. Eldridge, Amani Khalil, Laura Russo.

**Funding acquisition:** Laura Russo.

**Investigation:** Devon S. Eldridge, Amani Khalil, Laura Russo.

**Methodology:** Devon S. Eldridge, Amani Khalil, Laura Russo.

**Project administration:** John K. Moulton, Laura Russo.

**Resources:** Laura Russo.

**Supervision:** John K. Moulton, Laura Russo.

**Validation:** Devon S. Eldridge, Amani Khalil, John K. Moulton, Laura Russo.

**Visualization:** Devon S. Eldridge, Amani Khalil, Laura Russo.

**Writing – original draft:** Devon S. Eldridge, Amani Khalil.

**Writing – review & editing:** Devon S. Eldridge, Amani Khalil, John K. Moulton, Laura Russo.

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
