## [Decision Letter · Decision Letter 0]

9 May 2024

PONE-D-24-09528Do local and landscape context affect the attractiveness of flower gardens to bees?PLOS ONE

Dear Dr. Russo,

Thank you for submitting your manuscript to PLOS ONE. After careful consideration, we feel that it has merit but does not fully meet PLOS ONE’s publication criteria as it currently stands. Therefore, we invite you to submit a revised version of the manuscript that addresses the points raised during the review process.

The reviewers had widely split perspectives on your manuscript.  That said, reviewer 2 mirrored some of the concerns identified by reviewer 1, e.g. aspects of the model and areas where clarity was lacking.  Please carefully consider concerns raised by both reviewers; for any you disagree with, please clearly detail why in your response.

We look forward to receiving your revised manuscript.

Kind regards,

Dr. Janice L. Bossart

Academic Editor

PLOS ONE

“LR received funding from Bayer in 2019 ($5000 Feed A Bee).”

“We would like to thank the University of Tennessee (UT), the UT Institute of Agriculture, the UT Gardens, and the UT Ag Research and Education Centers, including the Plateau AgResearch Center, the Organic Crops Unit, and the Forest Research and Education Center, for allowing us to do research on their lands, especially K. Hoyt, B. Simpson, W. Hitch, H. Jones, J. Newburn, and W. Lively. We would also like to thank D. Matheson, S. Collins, and A. Murray for field assistance and N. Oldham for assisting with hoverfly identification. Thanks also go to S. Droege for lending his expertise in bee identification. We are also grateful to Bayer, for the Feed-A-Bee grant that funded our Feed-A-Bee research plots. Thank you to the Department of Ecology & Evolutionary Biology at the University of Tennessee for the support of this project.”

“LR received funding from Bayer in 2019 ($5000 Feed A Bee).”

5. Please remove your figures from within your manuscript file, leaving only the individual TIFF/EPS image files, uploaded separately. These will be automatically included in the reviewers’ PDF.

6. We note that Figure S1 in your submission contain [map/satellite] images which may be copyrighted. All PLOS content is published under the Creative Commons Attribution License (CC BY 4.0), which means that the manuscript, images, and Supporting Information files will be freely available online, and any third party is permitted to access, download, copy, distribute, and use these materials in any way, even commercially, with proper attribution. For these reasons, we cannot publish previously copyrighted maps or satellite images created using proprietary data, such as Google software (Google Maps, Street View, and Earth). For more information, see our copyright guidelines: http://journals.plos.org/plosone/s/licenses-and-copyright.

1. You may seek permission from the original copyright holder of Figure S1 to publish the content specifically under the CC BY 4.0 license. 

Reviewers' comments:

Reviewer's Responses to Questions

**Comments to the Author**

1. Is the manuscript technically sound, and do the data support the conclusions?

Reviewer #1: Partly

Reviewer #2: Yes

2. Has the statistical analysis been performed appropriately and rigorously? 

Reviewer #1: No

Reviewer #2: Yes

3. Have the authors made all data underlying the findings in their manuscript fully available?

Reviewer #1: Yes

Reviewer #2: Yes

4. Is the manuscript presented in an intelligible fashion and written in standard English?

Reviewer #1: Yes

Reviewer #2: Yes

5. Review Comments to the Author

Reviewer #1: This paper presents a study on the interplay between landscape and local-scale factors in shaping the attractiveness (in terms of plant-pollinator interactions and alpha diversity metrics) of floral bee gardens. This study considers two relevant spatial scales and investigates how important factors, such as resource availability and habitat amount, shape bee communities across human-dominated landscapes. The paper seems to be based on the master work of one of the authors, and thus, the limitations regarding sampling size (e.g., replicates per treatment) can be understood regarding the (probaly) limited humanpower and existing time constraints. The paper is well-written, structured, and organized. The authors have made the data available as declared.

Having said that, we have a number of major concerns regarding the (1) statistical analyses (families used and model assumptions, colinearity in the variables, definition of variables, model assembly, assumptions), (2) some of the methods, (3) some rationals regarding the conceptualisation of the study, and (4) the available datasets, which seem to have not been properly cleaned and are not sufficiently documented to reproduce and re-use the data (we have tried ourself and wasn’t able to reproduce the work). Thus, together, these points question the study as a whole and prevent to recommend the paper for publication in its current form. We think that there might be potential for publication, but this will require work.

I have added my comments below, which we hope will be useful for the authors

### Major comments: ###

- Statistical analyses:

o Correlations among predictors: The authors do not mention any collinearity check on their predictors, but this is a basic standard check. We’ve check the correlations of some of the predictors and they are quite high: floral display at the local scale vs. development; floral display at the local vs. plant richness at the local, plant richness at the local vs. development. The authors checked collinearity before building the models and checked the variance inflation factor (VIF) of the model. We guess the authors are familiar with it, but there is a large number of problems related to colinearity. This issue raises questions regarding the validity of the models.

o Statistical families of the models: The authors seem to have used Gaussian families for all their variables, despite working with count data (i.e., the assumptions of Gaussian do not necessarily work when you have count data, which is limited to countable values). While gaussian families are adequate when count data is faily large, it shouldn’t be used when the counts are “small” and there are zeros. This violates the assumption of constance variance. In your case, I think you should use “Poisson” family if there is no overdispersion, or using a negative binomial if overdispersion. This puts all models and results into question, as they are now.

o Model specifications and Table 1: The random effect structure specified in the method section and the one presented in table 1 do not match. In the methods, it is said that the random effects were site and sampling date nested in the site. However, in Table 1, it appears that garden type (or plot) was used in the random effect. It makes sense to use plot and sampling data as random effect to account for the lack of independence of these two factors, so perhaps there was an error in the method section.

Another problem is that there are five sampling rounds, but in Table 1, it appears that there are only four rounds. Please clarify this discrepancy.

- Other methods

o Ranking floral diversity and floral abundance: The authors used a scale of to 1-10 to quantify floral diversity and floral abundance. As presented in the Methods section, it is unclear what criteria were used to attribute the different scale values to the existing features. It feels that this is a purely subjective guess from the observer. It is also strange that no method based on coverage (E.g., Braun-blanquet scale or similar) was used. The authors should better explain how the assessment of their scale was done, how they dealt with the potential subjectivity of the observer, etc. As it is now, it leaves too many doubts on a variable that is key for the analyses.

o Estimation of floral abundance: the authors state that they counted inflorescens to assess floral abundance at both plot and local scales. They did not mention how they assessed single flowers (and they should mention it), but one can assume that they were counted individually. However, the major problem I see here is the lack of detail on the definition of inflorescences. Surprisingly, the authors did not use standard methods based on floral units (references). For example, how did you consider compound umbellas, compound capitulums, etc? A classification criteria for the different types of inflorescences and they way they were counted is necessary (e.g., see https://doi.org/10.1016/j.dib.2021.107243, https://doi.org/10.1038/s41559-018-0769-y for examples).

- Conceptual aspects:

o Definition of pollinator-friendly plants: The authors mention that they used pollinator-friendly plant species native to the study area. However, it is unclear what is used to define plant species as pollinator-friendly. Was this based on what nurseries and plant providers say? Is this based on actual research? This is crucial, as it has been found that what is reported by nurseries does not always reflect the reality.

o Functional ecology aspects: While the study stands as it is, I think it is difficult to justify why the authors did not use any functional metrics and quantify resources (and resource quality) solely using taxonomic metrics. In conclusion, the authors suggest increasing plant diversity and floral display, but we know that this is a poor recommendation without actually considering the value of plants for pollinators (one could add many grasses, plant varieties that do not provide rewards, or plants that have mismatches in their flowering season with the phenology of the pollinators). This can be achieved by integrating the functional traits of plants, which are quite widely available (TRY database, datapapers), such as blossom class, symmetry, flowering start and duration, growth form, and even nutritional metrics (e.g., sugar in pollen or nectar). Computing functional metrics could really better underpin your results and provide strong support for your conclusions, which at the moment feel vague and unspecific.

e.g.,: https://www.sciencedirect.com/science/article/pii/S2352340921005278,

https://doi.org/10.1002/2688-8319.12248, https://esajournals.onlinelibrary.wiley.com/doi/10.1002/ecy.3705,

- Problems with the provided data: The authors made the data available. However, we found a number of problems that prevented reuse and reproduced the results:

o In general, you should provide a read-me file (separated as a text document or as a new sheet in the excel), where you clearly explain the columns of both sheets. It is unclear what site and block are (one has four levels, the other has five). You should define all variables, their column names, units, etc. So an external user can understand everything

o Sheet “specimens’: There is something wrong with this sheet. For example, between rows 39-72, the plant species and plant long columns are wong: it says local and UT Arboretum. There is no information on the block, plot type, local-garden scale, and so on. No columns with full site names.

o Sheet for data analyses: You should add the development metric here, so it simplifies reproducing the analyses. The differences between the site and the block are unclear.

### Other comments: ###

INTRODUCTION

- The introduction does not provide sufficient information on key concepts. The authors should devote some text and references to them.

o Central place foragers: This is critical because we know that most bees forage within the proximity of their nest, with some exceptions. Thus, the local-scale factors may be significant. There is plenty of literature in that regard: 10.1002/ecy.3809, https://doi.org/10.1002/eap.2727, https://www.sciencedirect.com/science/article/abs/pii/S0065250415000367

o Filtering: The third hypothesis of this study refers to filtering. However, the term is not adequately introduced, even though there is literature on the topic for bees: https://onlinelibrary.wiley.com/doi/abs/10.1111/jbi.13772, 10.1098/rspb.2014.2849 , 10.1038/s41467-020-14496-6

o Attractiveness: The authors disclose the drivers of attractiveness without properly defining the term and provide an overview of what has been found to influence it. E.g.: https://doi.org/10.1111/1365-2435.12178, 10.7717/peerj.3066

o Honeybees and their effects on wildbees: It would be beneficial to explain the potential effects of managed pollinators, such as honeybees, on wild bees, especially where they are not native. It will also help understanding why some analyses were done excluding honeybees. https://peerj.com/articles/14699/ , https://esajournals.onlinelibrary.wiley.com/doi/10.1002/ecy.3939

- L94: The phrasing of the sentences is a bit unusual, and it is difficult to understand what was done.

METHODS

- Experimental design: How far were the sites? What criteria were used to select them? (permisions and special features??). Please clarify.

- L106: here it is said there are 5 sites, in the data, the column “site” has only 4 levels…

- L166: provide the version of ArcGIUS

- L167: provide the resolution of the NLCD

- L186: Provide the version of R. If you used Rstudio, provide citation and version as well

- You should provide a list of packages and versions used in the supplementary

- L188: Predictors were scaled only or centered.

- L193-204: This paragraph could be better written and explained; that is, the information could be presented in a more efficient way. For us, it is not clear that local- and garden-scale models were done separately. This can be said earlier, and is better specified. The random effects did not match those presented in Table 1. There was no mention of how assumptions for the model were checked (e.g., spatial autocorrelation).

DISCUSSION

- L326-330: The authors discuss evenness here, but this is a metric that has not been presented before in the introduction or methods. Please integrate this into the manuscript.

- The authors could add some information regarding the following aspects, which are not unfamiliar to them:

o Nutritional aspects of floral resources: In addition to discussing the effects of resources based on alpha-diversity metrics, the authors could make use of the existing literature (including the work of the senior author) to add depth to the discussion of how floral resources shape bee communities (niche partitioning, health, etc.), the role of nutritional landscapes in floral restoration, etc.

- It feels more references could be integrated, e.g.,: https://advaudo.weebly.com/uploads/1/2/6/8/126897510/current_opinion_in_insect_science_2015_vaudo.pdf, http://www.woodardlab.com/uploads/6/4/9/6/64969235/1-s2.0-s2214574517300615-main.pdf,

CONCLUSIONS

- L347: What does “healthy” mean here? This concept has not been previously addressed. The authors could use previous work to develop it more: https://www.cell.com/trends/ecology-evolution/fulltext/S0169-5347(21)00333-5?_returnURL=https://linkinghub.elsevier.com/retrieve/pii/S0169534721003335?showall=true,

- L349: What are the specific traits of plants? It is always the more the better?

- L359: What do you mean by “high quality”? I think you can develop this much more in the Introduction and Discussion, and as mentioned before, even include it by using plant traits.

TABLE 1

- Effect size: The authors report effect size, but do not explain what metric was used in the methods. If this is the estimate of the predictor, why is it not called an estimate? Otherwise, please define the effect size. The estimate with the standard error is provided in Table 1.

FIGURES:

- Figure 1: You can add a map and a better representation of the nested design.

- Figure 2: (b) explain what was used for the smooth line (lm?); (C) state again what the shaded bands indicate

- Figure 3: The points represent the raw data? Are the smooth lines based on the model predictions? If not, they should be based on your models, so that we can see how they fit the data. Again, explain how the lines are calculated (linear models?) and shaded bands, respectively.

- Figre 4: what was the stress of the NDMS? Maybe add it to the supplementary.

Reviewer #2: This is an interesting study and the methods are solid. The taxonomic resolution of bee species is especially impressive. Also, the discussion was well-organized and effective. I left many comments in the attached document. None are "make or break" for the paper, but some of them may (1) suggest easy modifications to the analysis (considering more radii than just 2000m, considering quasipoisson or negative binomial models) or (2) improve how the statistical methods are described.

6. PLOS authors have the option to publish the peer review history of their article (what does this mean?). If published, this will include your full peer review and any attached files.

Reviewer #1: No

Reviewer #2: No

---

## [Author Response · Author response to Decision Letter 0]

20 Jun 2024

PONE-D-24-09528

Do local and landscape context affect the attractiveness of flower gardens to bees?

PLOS ONE

Dear Dr. Russo,

Thank you for submitting your manuscript to PLOS ONE. After careful consideration, we feel that it has merit but does not fully meet PLOS ONE’s publication criteria as it currently stands. Therefore, we invite you to submit a revised version of the manuscript that addresses the points raised during the review process.

The reviewers had widely split perspectives on your manuscript. That said, reviewer 2 mirrored some of the concerns identified by reviewer 1, e.g. aspects of the model and areas where clarity was lacking. Please carefully consider concerns raised by both reviewers; for any you disagree with, please clearly detail why in your response.

We look forward to receiving your revised manuscript.

Kind regards,

Dr. Janice L. Bossart

Academic Editor

PLOS ONE

***We would like to thank the editor and two anonymous reviewers for their helpful feedback on our manuscript. We have significantly revised the manuscript according to these recommendations and feel that it is much improved. Some significant changes included updating and clarifying the methods, improving the attached data file, redoing all the statistical analyses with new model distributions and adding different landscape buffers to the supplemental materials, remaking figures 1 and 3, and adding 24 new references suggested by the reviewers. We hope that the revised manuscript meets your standards.

We address the reviewer comments on a point-by-point basis below. Our responses are demarcated with three asterisks (***).***

“LR received funding from Bayer in 2019 ($5000 Feed A Bee).”

***We amended the role of funder as advised. This now reads “Funding was provided by Bayer’s Feed-A-Bee grant to LR in 2019. The funders had no role in study design, data collection and analysis, decision to publish, or preparation of the manuscript.”***

“We would like to thank the University of Tennessee (UT), the UT Institute of Agriculture, the UT Gardens, and the UT Ag Research and Education Centers, including the Plateau AgResearch Center, the Organic Crops Unit, and the Forest Research and Education Center, for allowing us to do research on their lands, especially K. Hoyt, B. Simpson, W. Hitch, H. Jones, J. Newburn, and W. Lively. We would also like to thank D. Matheson, S. Collins, and A. Murray for field assistance and N. Oldham for assisting with hoverfly identification. Thanks also go to S. Droege for lending his expertise in bee identification. We are also grateful to Bayer, for the Feed-A-Bee grant that funded our Feed-A-Bee research plots. Thank you to the Department of Ecology & Evolutionary Biology at the University of Tennessee for the support of this project.”

“LR received funding from Bayer in 2019 ($5000 Feed A Bee).”

***We removed the statement from the acknowledgements section. It now reads “We would like to thank the University of Tennessee (UT), the UT Institute of Agriculture, the UT Gardens, and the UT Ag Research and Education Centers, including the Plateau AgResearch Center, the Organic Crops Unit, and the Forest Research and Education Center, for allowing us to do research on their lands, especially K. Hoyt, B. Simpson, W. Hitch, H. Jones, J. Newburn, and W. Lively. We would also like to thank D. Matheson, S. Collins, and A. Murray for field assistance and N. Oldham for assisting with hoverfly identification. Thanks also go to S. Droege for lending his expertise in bee identification. Thank you to the Department of Ecology & Evolutionary Biology at the University of Tennessee for the support of this project.”***

***All data used for the analysis are in the attached supporting file.***

5. Please remove your figures from within your manuscript file, leaving only the individual TIFF/EPS image files, uploaded separately. These will be automatically included in the reviewers’ PDF.

***We removed the figures from the manuscript file.***

6. We note that Figure S1 in your submission contain [map/satellite] images which may be copyrighted. All PLOS content is published under the Creative Commons Attribution License (CC BY 4.0), which means that the manuscript, images, and Supporting Information files will be freely available online, and any third party is permitted to access, download, copy, distribute, and use these materials in any way, even commercially, with proper attribution. For these reasons, we cannot publish previously copyrighted maps or satellite images created using proprietary data, such as Google software (Google Maps, Street View, and Earth). For more information, see our copyright guidelines: http://journals.plos.org/plosone/s/licenses-and-copyright.

1. You may seek permission from the original copyright holder of Figure S1 to publish the content specifically under the CC BY 4.0 license. 

***Based on Reviewer 1’s comments, we removed Fig. S1 and replaced Fig. 1 with a map-based figure, using the USGS EROS public domain maps listed above.***

***We added the captions for our supporting information tables and figures at the end of the revised manuscript.***

5. Review Comments to the Author

Reviewer #1: This paper presents a study on the interplay between landscape and local-scale factors in shaping the attractiveness (in terms of plant-pollinator interactions and alpha diversity metrics) of floral bee gardens. This study considers two relevant spatial scales and investigates how important factors, such as resource availability and habitat amount, shape bee communities across human-dominated landscapes. The paper seems to be based on the master work of one of the authors, and thus, the limitations regarding sampling size (e.g., replicates per treatment) can be understood regarding the (probaly) limited humanpower and existing time constraints. The paper is well-written, structured, and organized. The authors have made the data available as declared.

***We thank the reviewer for their thorough and helpful comments. We have addressed your comments on a point-by-point basis, demarcated by three asterisks (***).***

Having said that, we have a number of major concerns regarding the (1) statistical analyses (families used and model assumptions, colinearity in the variables, definition of variables, model assembly, assumptions), (2) some of the methods, (3) some rationals regarding the conceptualisation of the study, and (4) the available datasets, which seem to have not been properly cleaned and are not sufficiently documented to reproduce and re-use the data (we have tried ourself and wasn’t able to reproduce the work). Thus, together, these points question the study as a whole and prevent to recommend the paper for publication in its current form. We think that there might be potential for publication, but this will require work.

I have added my comments below, which we hope will be useful for the authors

### Major comments: ###

- Statistical analyses:

o Correlations among predictors: The authors do not mention any collinearity check on their predictors, but this is a basic standard check. We’ve check the correlations of some of the predictors and they are quite high: floral display at the local scale vs. development; floral display at the local vs. plant richness at the local, plant richness at the local vs. development. The authors checked collinearity before building the models and checked the variance inflation factor (VIF) of the model. We guess the authors are familiar with it, but there is a large number of problems related to colinearity. This issue raises questions regarding the validity of the models.

***To address this concern, we ran separate models for the plant diversity and abundance terms, and then used AIC to select the better model.***

o Statistical families of the models: The authors seem to have used Gaussian families for all their variables, despite working with count data (i.e., the assumptions of Gaussian do not necessarily work when you have count data, which is limited to countable values). While gaussian families are adequate when count data is faily large, it shouldn’t be used when the counts are “small” and there are zeros. This violates the assumption of constance variance. In your case, I think you should use “Poisson” family if there is no overdispersion, or using a negative binomial if overdispersion. This puts all models and results into question, as they are now.

***The Poisson models were overdispersed. In the revised manuscript, we now use glmmTMB to build negative binomial models.***

o Model specifications and Table 1: The random effect structure specified in the method section and the one presented in table 1 do not match. In the methods, it is said that the random effects were site and sampling date nested in the site. However, in Table 1, it appears that garden type

---

## [Editor Report · Decision Letter 1]

11 Jul 2024

PONE-D-24-09528R1Do local and landscape context affect the attractiveness of flower gardens to bees?PLOS ONE

Dear Dr. Russo, Thank you for submitting your manuscript to PLOS ONE. After careful consideration, we feel that it has merit but does not fully meet PLOS ONE’s publication criteria as it currently stands. Therefore, we invite you to submit a revised version of the manuscript that addresses the points raised during the review process.

We look forward to receiving your revised manuscript.

Kind regards,

Dr. Janice L. Bossart

Academic Editor

PLOS ONE

Journal Requirements:

**Additional Editor Comments:**

The revised version has corrected most of the problems identified by reviewers.  Thank you for your efforts to address their concerns.  Below I list remaining issues that need attention.

Line 124.  Missing 'the'.  Three of the...

Line 439.  Missing word.  There 'was' ?

Line 728.  Please spell out linear model(s) so it's not confused with 1m, and either make model plural or add an 'a' before linear.

Table 1.  Delete the extraneous 's' at the end.

Figure S1.  The legend struck me as a bit odd given 'Bees' aren't an Order.  Maybe Hymenoptera-Bees and Hymenoptera-Non Bees?  Or maybe you can think of some other way that all terms fit.

Figure 3C.  Given the overlap of confidence intervals of 'Mixed' and at least 'Organic', I'm not sure I buy the interpretation that both 'Forage' and 'Mixed' had higher Shannon Diversity than all others.

--Mismatch between Figure 3 and Figure 3 caption.  Letter designations in the Figure caption don't match letter designations associated with plots in the actual figure.  Also, there is no plot of local bee abundance and land development (I'm guessing it's the caption that's incorrect given what is stated in the Results section).  

--Mismatch between Table 1 and text in the Methods.  Is sampling round nested within Site or nested within Plot?  The Table says Plot, the text says Site (Line 273).  Regardless of which is correct, the text would read better if you delete 'identity'.

--Loose use of 'diversity' throughout.  Given diversity can be used in multiple ways, please don't use it when you are referring specifically to species richness of either bees or flowers.  Otherwise, readers can't be sure whether you mean diversity, in terms of both richness and evenness, or simply richness.  Please carefully go through the manuscript, including figures and tables, and replace your use of 'diversity' with 'richness' anywhere you are explicitly referring to richness.

--Be consistent across Text, Figures, Tables, etc. including supplemental.  For example, Figure S2 uses Shannon Diversity and Figure 2C uses Shannon Index.  I didn't check every use, but Shannon diversity is often used in the text so perhaps that would be the way to go throughout.  Also, Figure S2C is unnecessary given it is identical to Figure 2C.  

--Rarefaction & Coverage methods. There is insufficient information in this section of the manuscript, e.g. approx. 256-267, to understand data shown in the resultant Figures.  iNEXT calculates Hill numbers, which at least needs to be pointed out as these are different calculations than other methods use.  Also, your figures don't correspond with figures generated by iNEXT for sample coverage or completeness, e.g. I'm expecting coverage or completeness to be an axis label.  No doubt I'm missing something, but if I am then others will likely also be confused.  Please add a bit of text to clarify how coverage was estimated and depicted.

---

## [Author Response · Author response to Decision Letter 1]

25 Jul 2024

PONE-D-24-09528R1

Do local and landscape context affect the attractiveness of flower gardens to bees?

PLOS ONE

Dear Dr. Russo,

Thank you for submitting your manuscript to PLOS ONE. After careful consideration, we feel that it has merit but does not fully meet PLOS ONE’s publication criteria as it currently stands. Therefore, we invite you to submit a revised version of the manuscript that addresses the points raised during the review process.

We look forward to receiving your revised manuscript.

Kind regards,

Dr. Janice L. Bossart

Academic Editor

PLOS ONE

***We would like to thank the editors for the helpful comments. We have addressed all the comments on a point by point basis below, demarcated with three asterisks.***

Journal Requirements:

Additional Editor Comments:

The revised version has corrected most of the problems identified by reviewers. Thank you for your efforts to address their concerns. Below I list remaining issues that need attention.

**Thank you for these helpful comments!***

Line 124. Missing 'the'. Three of the...

***Fixed.***

Line 439. Missing word. There 'was' ?

***Fixed.***

Line 728. Please spell out linear model(s) so it's not confused with 1m, and either make model plural or add an 'a' before linear.

***We changed this to “The lines are drawn with the function “geom_smooth(method = lm)” in the package ggplot2...” to clarify this sentence.***

Table 1. Delete the extraneous 's' at the end.

***Fixed.***

Figure S1. The legend struck me as a bit odd given 'Bees' aren't an Order. Maybe Hymenoptera-Bees and Hymenoptera-Non Bees? Or maybe you can think of some other way that all terms fit.

***We changed “Order” to “Taxa” in this figure to correct for the different categories.***

Figure 3C. Given the overlap of confidence intervals of 'Mixed' and at least 'Organic', I'm not sure I buy the interpretation that both 'Forage' and 'Mixed' had higher Shannon Diversity than all others.

***For the supplemental figure showing the results of the rarefaction analyses, we realized it should be labeled as Figure S2, instead of S3, so we corrected that. The Forage and Mixed had higher Shannon Diversity, but only when honeybees are included, whereas the figure in the main text shows the honeybees removed, so we clarified this in text. We also added a new supplemental table with the raw data output from the rarefaction analysis showing the estimated diversity and its confidence intervals, and the sample coverage for the sites and surveys.***

--Mismatch between Figure 3 and Figure 3 caption. Letter designations in the Figure caption don't match letter designations associated with plots in the actual figure. Also, there is no plot of local bee abundance and land development (I'm guessing it's the caption that's incorrect given what is stated in the Results section). 

***Thank you for catching that! We corrected the figure caption to match the figure.*** 

--Mismatch between Table 1 and text in the Methods. Is sampling round nested within Site or nested within Plot? The Table says Plot, the text says Site (Line 273). Regardless of which is correct, the text would read better if you delete 'identity'.

***The table was correct, so we modified the main text to match. We deleted “identity” in the text.***

--Loose use of 'diversity' throughout. Given diversity can be used in multiple ways, please don't use it when you are referring specifically to species richness of either bees or flowers. Otherwise, readers can't be sure whether you mean diversity, in terms of both richness and evenness, or simply richness. Please carefully go through the manuscript, including figures and tables, and replace your use of 'diversity' with 'richness' anywhere you are explicitly referring to richness.

***Thank you for this, we have revised the text throughout, and checked the figures in the main text and supplement. We include “diversity” rather than “species richness” under the following circumstances: 1) for bee diversity when we are referring both to species richness and Shannon diversity, 2) for the plant diversity where we ranked floral diversity on a scale of 1-10 rather than quantifying species richness of the flowering plants. In addition to these edits, we reviewed our use of floral display throughout to make it more consistent.***

--Be consistent across Text, Figures, Tables, etc. including supplemental. For example, Figure S2 uses Shannon Diversity and Figure 2C uses Shannon Index. I didn't check every use, but Shannon diversity is often used in the text so perhaps that would be the way to go throughout. Also, Figure S2C is unnecessary given it is identical to Figure 2C. 

***We changed Shannon Index to Shannon Diversity throughout and checked the figures as well. We agree that is the more appropriate term. We removed Figure S2C (it was replaced with sample coverage graphs).***

--Rarefaction & Coverage methods. There is insufficient information in this section of the manuscript, e.g. approx. 256-267, to understand data shown in the resultant Figures. iNEXT calculates Hill numbers, which at least needs to be pointed out as these are different calculations than other methods use. Also, your figures don't correspond with figures generated by iNEXT for sample coverage or completeness, e.g. I'm expecting coverage or completeness to be an axis label. No doubt I'm missing something, but if I am then others will likely also be confused. Please add a bit of text to clarify how coverage was estimated and depicted.

***Thank you for pointing out this omission. We added detail about the methods used (and Hill numbers) to the methods. We also add new graphs to the supplemental figure to show the sampling completeness of the surveys and sites.***

---

## [Editor Report · Decision Letter 2]

5 Aug 2024

Do local and landscape context affect the attractiveness of flower gardens to bees?

PONE-D-24-09528R2

Dear Dr. Russo,

We’re pleased to inform you that your manuscript has been judged scientifically suitable for publication and will be formally accepted for publication once it meets all outstanding technical requirements.  Congratulations!

There are two minor issues that first need to be corrected.  I have alerted the journal editor that these should be corrected before publication:

Line 754.  Figure 3 Caption.  Move (E) forward to follow 'richness' versus where you currently have it.Figure S2 Caption, line 5. Should be '...abundant species affects...', i.e. affect should be plural.

Kind regards,

Dr. Janice L. Bossart

Academic Editor

PLOS ONE
---

## [Editor Report · Acceptance letter]

8 Aug 2024

PONE-D-24-09528R2 

PLOS ONE

Dear Dr. Russo, 

I'm pleased to inform you that your manuscript has been deemed suitable for publication in PLOS ONE. Congratulations! Your manuscript is now being handed over to our production team.

Kind regards, 

on behalf of

Dr. Janice L. Bossart 

Academic Editor

PLOS ONE